# The 'fragile' generation: Understanding parents' attitudes towards young people's mental health within the Chinese sociocultural context

**Ziqi Huang**⊙°*, **David M. Frost**⊙°

Social Research Institute, Thomas Coram Research Unit, University College London, London, United Kingdom

☙ The authors contributed equally to this work.
* ziqi.huang.19@ucl.ac.uk

## Abstract

This study explored the interactions between parents' attitudes towards young people's mental health and the Chinese sociocultural context, using labelling theory and the ecological systems model. An online qualitative survey involving a sample of 126 Chinese parents was conducted. Through sequential content analysis and thematic analysis, results illustrated that Confucianist family relationship patterns, the one-child policy, anomie and public stigma all contributed to parents' paradoxical attitudes. Parents with diagnosed children reported less stereotypical attitudes and more detailed reflections on causal attributions regarding mental health difficulties. However, parents commonly reported internalised stigma and helplessness in terms of coping strategies due to common constraining sociocultural factors. Findings highlighted that the label of mental health difficulties remained a stigmatising concept that categorised young people as abnormal even when parents were aware of the existence of stigma. Parents also showed limited incentives to directly support children's mental health due to their dependency on young people's future success for caring. Implications for improving parental support and public education surrounding mental health are discussed.

## Introduction

During the past decade, young people's mental health has attracted growing public attention in China with increasing rates of various mental health difficulties (MHD), including below diagnostic-level symptoms, mental disorders, self-harm behaviours, suicide ideations and suicide attempts [1]. According to the World Health Organization (WHO), it appears that although the nation has experienced rapid economic growth and higher quality of life over the past few decades, young people's mental health has curiously worsened [2,3]. At the time of this writing, researchers were not aware of any official statistics or publicly available nationally representative datasets

**Data availability statement:** The data used in this study contain potentially identifiable and sensitive information, as participants were invited to discuss their children's age, gender, mental health difficulties, and personal life stories (which may have included 3rd party identifying information). According to the information sheets and consent forms agreed by the participants, as well as the ethics protocol, the data cannot be shared with any third party or made publicly available. Specifically, participants consented to the following condition: "Only my supervisor and I will have access to the data, which will be kept until September 2023 when the project is complete (see Page 2)." Furthermore, the ethics protocol states: "All data will be destroyed when the project ends." Due to these ethical restrictions, we are unable to share the data once the manuscript has been published. If there are any questions about these ethical restrictions on public access to the data, they can be directed to: ioe.researchethics@ucl.ac.uk.

**Funding:** The authors received no specific funding for this work.

**Competing interests:** The authors have declared that no competing interests exist.

regarding the prevalence of young people's MHD in China. However, various studies confirm the increasing concern regarding this issue [4, 5, 6, 7]. For example, two recent systematic reviews report that 28.4% of Chinese university students have experienced depressive symptoms [6], while 28% of graduate students have illustrated symptoms of mental distress [7]. Suicide had become the fourth leading cause of death for people aged 15–29 by 2019 [2].

The potential role of parental attitudes in shaping young people's mental health has yet to be examined in the context of the increasing prevalence of MHD among young people in China. Research has emphasised that parents' attitudes can largely affect whether or when young people seek help for MHD, as well as their continuance of treatment once started [8, 9, 10]. Positive attitudes, such as encouragement of help-seeking, act as a protective factor against young people' MHD, while negative attitudes, such as criticism or denial of the problem, can prohibit young people from receiving treatment [11,12]. In China, the family plays a central role in the organisation of society and functions as the main source of support for young people, even after the transition into early adulthood [13,14]. Family communication also plays a fundamental role in young people's socialisation and development of positive mental health [15]. Parental influences may thus continue to have considerable impacts on young adults' mental health, however research has yet to interrogate the degree to which parental factors contribute to increasing MHD among Chinese young people. Therefore, the current study focuses on parents' expectations, impressions, and potential stigmatisation of MHD with the ongoing mental health crisis, as these may all play a role in shaping young people's mental health.

### How does the Chinese sociocultural context shape parental influence on young people's MHD?

While parents' attitudes toward young people's mental health are embedded within multiple layers of contextual factors [16,17], research in China has yet to fully consider the ways in which recent social changes and cultural norms shape parents' attitudes toward their children's MHD. As Bronfenbrenner's [18] ecological systems model argues, individual wellbeing is shaped by multi-level contextual factors such as family, community, social institutions and cultural values. Thus, to better understand contextual factors shaping young people's mental health, researchers need to understand their parents' attitudes and experiences, which are embedded in micro-, meso-, and macro-level systems involving family relationships, Chinese social policies, anomie, the collectivistic culture, as well as the interconnections among these levels. We provide Fig 1 as an illustration of these yet to be examined contextual factors, and expand upon each in the sections that follow.

**Confucianist family values may magnify the importance of parental influence.** Traditional Chinese families usually follow the pattern of Confucianist cultural values [1,15,19]. In a Confucianist family, family relationships exist in hierarchal patterns where children are expected to obey their parents, and mothers are expected to obey their husbands [19]. Health care decisions are often made collectively by the whole family with these hierarchal relationships in mind [19].

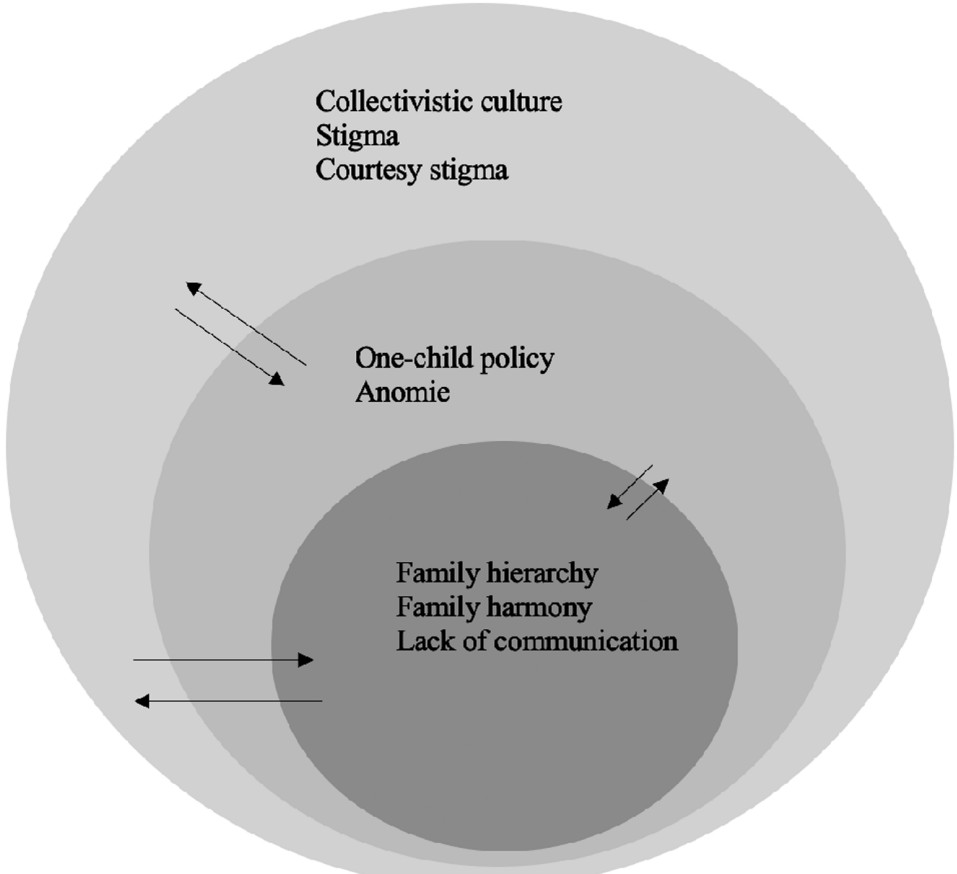

**Fig 1. The sociocultural context within which Chinese emerging adults live.**

Therefore, young people's help-seeking decisions regarding mental health can still be considerably influenced by their parents' attitudes, even as young adults. Additionally, Confucianism stresses harmony and the overall good of the family [20], so young people may conceal their MHD to avoid conflicts or embarrassment [13]. Traditional Chinese families also tend to keep private concerns such as MHD within the family to preserve reputation of the whole family [15,20]. This arguably leads to fewer opportunities for parents to intervene in helping their children cope with or seek treatment for MHD. Even if young people reveal their MHD, whether they will receive treatment often depends on parents' attitudes [8].

**The one-child policy may exacerbate anomie.** While traditional family values have persisted, the Chinese family structure has been changed in the past few decades by the one-child policy [21]. The one-child policy is a birth control policy that was officially announced in China in 1980 to address rapid population growth at the time, which stipulated that one couple could only have one child [21]. Although this policy has recently expired, for many young adults around university-age, they are still, or have been, the only child in the family. This means that a significant part of traditional family connections and support, that is, siblings, have been absent for these younger generations. Frequently, the one-child policy functions as an 'informal contract' between parents and the state: while the state restricts the number of children parents can have, they are promised that their one child will be of high quality and pay back their parents in the future [1]. As a result, parents tend to make great sacrifices for their child's education, hoping to receive rewards from their child in the future [21]. However, because there is no clear definition of success [13], young people may be pressured

to keep making progress without a set destination. Such pressure to continuously pursue increasingly demanding goals corresponds with Durkheim's [22] concept of anomie, which argues that individuals relentlessly chasing after better achievements and unknown pleasures may eventually lead to suicidal ideations, because no one can ever achieve happiness when the goal itself is limitless. A recent study shows that high parental control is significantly related to young people's heightened perceptions of anomie, which in turn leads to higher suicidality [23].

**Collectivistic culture may increase the importance of stigma.** Although group-level categorisation of individualistic vs. collectivistic cultures has often been criticised for being over-simplifying [24], the characteristics of collectivistic cultures still largely apply to Chinese society regarding two aspects: emphasis on interpersonal relationships and the central role of family in individual life [25]. Compared to those in the Western context, Chinese young people tend to be more influenced by interpersonal interactions and others' opinions [26]. Surveys confirm that a primary barrier to young people's help-seeking is their fear of prejudice and discrimination [9,27]. Stigmatisation of MHD still widely exists in Chinese society. For example, previous studies have identified that older generations and caregivers may see people struggling with MHD as lazy and attention-seeking [28,29]. Peers and family members may perceive them as pathetic, abnormal or dangerous, leading to social isolation [8,30].

**The role of social stigma in shaping parental attitudes toward young people's mental health.** Although classically defined [31] as 'an attribute that is deeply discrediting', stigma is essentially a process through which individuals are distinguished by a certain label, socially isolated, afforded disadvantaged social status, and marginalised as non-normative groups in society [3 2]. Mental illness can be a label that becomes public once someone is known to use mental health services [33,34]. Additionally, family members of people who have MHD may experience what Goffman [31] calls courtesy stigma, that is, extended public stigma and loss of social status because they are known to associate with stigmatised individuals. For example, parents who have children with MHD may internalise assumptions such as the family is problematic, or that they are bad parents [33]. Research confirms that over 60% of family members in China report trying to conceal their relationships with people who have mental disorders due to a sense of shame, which possibly causes reduced help-seeking behaviours [35]. However, research has yet to investigate the extent to which stigma in general, and parental courtesy stigma more specifically, play a role in shaping Chinese parents' attitudes toward young people's mental health.

## The current study

While current evidence highlights the existence of stigmatising attitudes towards MHD within Chinese communities [8,28–30], this study was the first to explore parents' attitudes towards young people's mental health within the current Chinese sociocultural context. Furthermore, it explored how parental attitudes could shape and interact with parent's daily practices in terms of supporting their adult children's mental health. In light of the increasing societal attention being paid to MHD among younger generations in China, the current study aimed to understand contextual factors shaping parental attitudes in general, and did not solely focus on the attitudes of parents who had children with MHD. Additionally, the study focused on how parents negotiated stigmatisation and support for children's mental health, regardless of their children's current mental health conditions. Specifically, the current study investigated the following research questions: (1). What are Chinese parents' attitudes towards young people's mental health? (2). How does the sociocultural context shape parents' attitudes towards young people's mental health? (3). How do Chinese parents cope with young people's MHD both emotionally and practically?

## Method

Data collection was conducted through an online qualitative survey. Given the sensitive nature of the research, this method was chosen because it allowed for researchers to collect data about parents' attitudes in a neutral and anonymous way. As mentioned above, the presence of the researcher as an outsider to the family may have enhanced feelings

of discomfort or shame when discussing the MHD of family members. Thus, an open-ended questionnaire was used instead of interviews in order to encourage participants to freely share their own perspectives along with information about their children's MHD by enabling them to answer questions in their own homes and fully anonymously.

### Recruitment and participants

This study targeted Chinese parents whose children were 18–25 years old. The age span of emerging adulthood (18–25) is an especially sensitive life period when young people are experiencing various major life changes regarding work, love and worldviews, transitioning from childhood to complete adulthood [36]. Recent statistics show that emerging adults aged 18–25 are among the most vulnerable age groups in terms of mental health risks [2]. Paradoxically, while emerging adults generally illustrate reduced risks of depression and increased mental well-being, they also experience highest risks of major psychopathology like schizophrenia and bipolar disorders [37]. Therefore, the age span of 18–25 may show complex and diverse patterns of mental health development beyond simply becoming better or worse. Additionally, with young people struggling to become independent but still partially attached to their parents for practical support [36,37], this life period might present different challenges to traditional hierarchal parent-child relationships in China, potentially affecting how health care decision-making is negotiated within families.

Participants were recruited during February 15, 2023 – March 15, 2023 through purposive sampling. Links to the survey were published on Chinese social media platforms including WeChat and Rednote. Given that there had previously been limited research regarding Chinese parents' attitudes about mental health since the pandemic, these two platforms were chosen due to their popularity in China and the potential to collect qualitatively rich data. Snowball sampling was also used to increase the diversity of the sample beyond those parents using these platforms. Specifically, the researchers personally contacted potential participants to ask if they were willing to take part in this study and participants were encouraged to invite their acquaintances to complete the questionnaire.

In total, 378 people responded to calls for participation, and 287 people provided their consent to participate. Of these responses, 161 (56.1%) were not considered eligible because they did not meet the eligibility criteria, had extensive missing data or responded after the data collection deadline. After these exclusions, the questionnaire received 126 valid responses. All participants identified themselves as Chinese and completed the questionnaire in Mandarin. Participants' ages ranged from 28 to 56, with a mean age of 47 years old. 63 (50.0%) participants were female, 61 (48.4%) were male and 2 (1.6%) preferred not to say. Only 5 (4.0%) participants were from rural areas, while the rest 121 (96.0%) were urban residents. Findings discussed below are therefore not generalisable across rural/urban areas in China. As some have found education level to be significantly correlated with stigma about mental health [38], this study also asked about participants' education level. 74 (58.7%) participants reported having a bachelor's degree, while 22 (17.5%) reported having a master's degree or above and 18 (14.3%) held a high school diploma or below.

### Questionnaire development

A questionnaire was developed containing structured prompts to elicit data relevant to the research questions, which included different branches to distinguish between parents with/without children diagnosed with mental illness, or those who believed that there existed/did not exist social stigma around mental health. To avoid ambivalence about the definition of "mental health difficulties", this term was sometimes used interchangeably with other commonly used terms in Chinese such as "mental health problems" in the questionnaire, so that it was easier for parents to understand. The questionnaire was first constructed via Qualtrics in English, then translated and adapted into a Mandarin version before publishing. It contained questions structured around the following topics: Parent's impressions of young people with MHD; their perceptions of parents' role in young people's mental health; what sociocultural factors they believe affect young people's mental health; and how they cope with young people's MHD. A copy of the questionnaire is included in the supplemental materials.

**Ethics**

This study received ethical approval from UCL Institute of Education Research Ethics Committee (IOE REC) on January 16, 2023. Informed consent was obtained from all participants, as participants were required to confirm that they were over 18 years old and that they agreed to participate in this study before proceeding to the questionnaire.

**Data analysis**

Thematic analysis was conducted to identify, analyse and report themes in the data [39] through an integrated interpretative and explanatory approach. Themes were generated following Braun and Clarke's [39] six-step procedure, including familiarisation with the data, generalising initial codes, searching for themes, reviewing themes, defining/naming themes and producing the report. The coding was led by the first author, who was Chinese and was therefore familiar with phrases used by the participants as well as the Chinese context. The initial codes were discussed with the second author, a White American social scientist with expertise in research on stigma and mental health and refined further though additional application to the data. A detailed codebook was then created with conceptual and operational definitions of each code, along with coding application rules and examples of coded data. Codes were refined, discarded and reapplied during this process, with emergent themes being discussed after the initial coding process, and refined and reapplied after initial application. Multiple meetings in this iterative process were devoted to the discussion of inconsistencies in the coding process, refining coding definitions and rules for their application in order to strive toward coding consistency and agreement in the development of themes. Four themes were finalised through this recursive process. Those regarding parents' personal attitudes and stigma were derived from an inductive approach striving for a comprehensive portrait of the data relevant to the first research question, while those regarding sociocultural influences followed a more deductive or theoretically driven approach in line with the study's second research question. It was taken into consideration that evaluating the data's importance independently based on the frequency of reoccurrence can be insufficient [40]. Therefore, some themes were generated based on the content's potential meaningfulness rather than numeric frequency. Exemplar quotations as evidence for analytic claims were selected from the original data in Mandarin and translated to English for the purposes of presentation in this manuscript.

Among the data we collected, several terms, including mental disorders, mental illness, mental health difficulties and psychological issues, were sometimes used interchangeably by participants. These terms are kept in the results presented below because we believe that they reflect how MHD are interpreted by parents in lay conversations. However, when describing and discussing the results, we use MHD as an umbrella term referring to any below diagnostic-level symptoms, mental illnesses, self-harm and suicide.

# Results

Four themes were identified: *contrasting accounts of attitudes and perceived stigma; perceived unescapable anomie; the underlying logic of Confucianism ideology*; and *attitudes of parents with and without diagnosed children*. The first theme related primarily to the first research question and reflected how parents held complex and contradictory attitudes towards young people's mental health. The second and third themes related primarily to the second and third research questions. Theme 2 reflected how various sociocultural factors contributed to parents' perceived anomie in contemporary Chinese society and their corresponding attitudes towards this emerging challenge to young people's mental health. Theme 3 reflected how Confucianist ideology functioned as a psychological mechanism that justified parents' accounts in the first two themes. Finally, theme 4 related primarily to the third research question pertaining to differences and similarities in attitudes present in parents who had children with diagnosed mental illness and those who did not.

**Theme 1: Contrasting accounts of impressions and perceived stigma**

**General impressions.** When participants were asked about why some young people might experience MHD, as well as their impressions of these young people, the word 'fragile' (脆弱) emerged as a key characteristic that was repeated throughout the dataset. The term held various meanings, such as young people are spoiled, mentally weak and incapable of dealing with stress, or that they lack the ability to manage life difficulties independently. Generally, it implied a sense of weakness, incapability or incompetence. For example, one participant commented:

> They are incapable of fitting into society because they are too fragile and sensitive… they do not have the ability to manage life difficulties and tend to escape from the reality in these situations. Therefore, negative emotions begin to accumulate, leading to desperation and hopelessness. *(Female, 50 years old, 22-year-old child, high school degree)*

Another participant noted:

> They are mentally fragile and dependent on others, therefore lack the capability to address problems by themselves *(Female, 22-year-old child, age and education level not reported)*.

Particularly, the one-child policy was identified by some participants as a crucial factor contributing to young people's 'fragileness':

> …the one-child policy has created a generation that is selfish and irresponsible *(Female, 48 years old, 20-year-old child, bachelor's degree)*.

> Due to the one-child policy, many parents today spoil their children, which is why those young people are too fragile. They have never experienced any difficulties or setbacks *(Male, 50 years old, 19-year-old child, bachelor's degree)*.

In addition to being 'fragile', poor mental health was often translated into defective personalities. For instance, some believed that these young people can be too quiet or self-isolated:

> They are sensitive and insecure… often think too much, yet do not express those ideas *(Female, 45 years old, 22-year-old child, high school degree)*.

> They are not good at communicating with others and are always self-enclosed. Those kids are lonely and antisocial *(Male, 49 years old, 21-year-old child, bachelor's degree)*.

On the other hand, others viewed them as emotionally unstable and potentially dangerous:

> […] they are impulsive and filled with radical emotions. When faced with difficulties, they are likely to conduct extreme behaviours *(Female, 47 years old, 21-year-old child, master's degree or above)*.

> They are mentally weak and are likely to be extreme or radical… overall, emotionally unstable. It's difficult to understand them *(Male, 48 years old, 20-year-old child, bachelor's degree)*.

Generally, participants' rhetoric reflected that MHD was often seen as related to abnormality:

> … the rapid growth of economics and inequality has resulted in the abnormality of young people's state of mind *(Male, 49 years old, 20-year-old child, bachelor's degree)*.

… young people with mental disorders may appear 'normal' most of the time, but through daily contact, they'll gradually show abnormality at some aspects *(Female, 45 years old, 22-year-old child, bachelor's degree)*.

To explain this abnormality, the family was frequently mentioned as a common cause of young people's MHD. As mentioned previously, some participants attributed parents' spoiling as the main cause of young people's 'fragileness'. Others noted that:

…family is the primary cause of mental health difficulties. The harm caused by an unhappy childhood and their parents' tragic marriage will be lifelong, I'm afraid. These children will lack the capability of fitting into the society *(Female, 47 years old, 21-year-old child, high school degree)*.

…those parents didn't provide enough support, love or security for their children. This can lead to young people conducting extreme behaviours *(Female, 48 years old, 18-year-old child, master's degree or above)*.

**Perceived stigma.**  Despite the generally negative impressions of young people's MHD, most participants agreed that there were stereotypes and social stigma about young people's mental health. The following examples illustrate the most consistently identified stereotypes:

Many people are biased… they think young people with mental health problems are either dangerous or just being dramatic because they are incapable of dealing with stress. These people simply lack empathy and ignore others' feelings *(Female, 45 years old, a 22-year-old child, bachelor's degree)*.

There exists stigmatisation such as 'young people are all spoiled, fragile, lazy…' *(Female, 48 years old, 18-year-old child, master's degree or above)*.

Some believe that young people today are privileged and are supposed to be happy. Mental illness is all about them being dramatic. They have never really entered the child's world. They have never understood young people's struggle and helplessness *(Male, 48 years old, 18-year-old child, master's degree or above)*.

The very phrase 'mental illness' is a label that leads to loss of social status, loss of job opportunities and others' prejudice *(Male, 49 years old, 20-year-old child, bachelor's degree)*.

These accounts of perceived stigma showed an overall contradictory perception with participants' own attitudes towards young people's mental health. While many participants recognised existing social stigma regarding mental health, they may paradoxically illustrate stigmatising attitudes regarding causal attributions and abnormality when the word stigma is not explicitly mentioned.

### Theme 2: Perceived unescapable anomie

**Anomie observed by participants.**  Anomie refers to a state of the society where individuals continuously pursue limitless goals but can never achieve happiness, finally losing their desire to live due to the impossibility of self-realisation [21]. As mentioned previously, many participants identified the increasing stress due to intense competitions for education and work opportunities as a significant contributor to young people's poor mental health:

Due to the education policies, the pressure of schoolwork has been increasing and the competitions are more intense than ever, which may affect young people's mental health *(Male, 51 years old, 22-year-old child, bachelor's degree)*.

Over the past few decades, money has almost become the only standard of 'success' in China. Both parents and children are pursuing this enforced ideology rather than what they really want. Parents are being strict on young people but

do not show enough respect for them… people have lost a self-identity *(Male, 51 years old, master's degree or above, child age not reported)*.

This was viewed as enhanced by parents' high expectations of their children. Especially, under the one-child policy, parents' expectations could be overwhelming:

…some parents have unreasonably high expectations of their children. It can be too much, especially for young people with a strong sense of responsibility. Such expectations will eventually turn into young people's feelings of guilt, self-blame and hopelessness *(Male, 50 years old, 20-year-old child, master's degree or above)*.

Young people today are mostly the only child in the family. Parents tend to have high expectations of them becoming successful. However, they are still kids, after all. They are bearing too much pressure from the older generation yet have no way to release the stress… *(Female, 48 years old, 18-year-old child, bachelor's degree)*.

**Lack of strategies to cope with anomie.** Although participants recognised the negative impacts of parents' high expectations on young people's mental health, they did not illustrate an apparent incentive to change the situation. Among 126 participants, only 2 mentioned reducing parents' expectations of success and placing less pressure on young people as a strategy to support young people's mental health. Instead, most participants expected young people to cope with the stress, control their emotions and 'keep fighting':

…it's important to give young people opportunities to release their emotions and make them aware that the world is worth keeping fighting for, so that they know what to do when faced with difficulties *(Female, 45 years old, 18-year-old child, bachelor's degree)*.

I believe that over-protecting children is bad for them. Problems result from their incapability of dealing with stress or being inexperienced about failures… *(Male, 45 years old, 18-year-old child, bachelor's degree)*.

During this process, parents largely perceived their role as providing indirect support. The most reported strategy was shaping young people's values through daily conversations. Others included observation, listening or giving advice for life difficulties:

… I try to spot potential threats to her mental health through daily conversations. It's important to help young people establish positive values and be optimistic about life difficulties *(Male, 49 years old, 21-year-old child, bachelor's degree)*.

I usually observe him and ask him if he's encountered any difficulties when he seems to be in a bad state. I'd also be his listener and tell him what attitudes or strategies he should employ to address the problem *(Male, 55 years old, 23-year-old child, bachelor's degree)*.

We rarely talk about mental health directly. I would use news or examples of other kids to start a conversation and try to shape his values and opinions *(Female, 48 years old, 18-year-old child, master's degree or above)*.

Generally, parents seemed to lack effective ways to directly reduce young people's stress. Instead, they preferred strategies that helped young people develop an optimistic attitude and manage negative emotions. Parents' high expectations were not discussed as something that needed to be changed. Rather, the pressure of anomie was implicitly illustrated as something that young people should try to accept with a positive attitude while they keep striving for a successful future.

### Theme 3: The underlying logic of confucianism ideology

**Family hierarchy: The attribution of responsibility.** As mentioned above, traditional Confucianism in Chinese culture emphasises filial piety and family hierarchy, both centring around children's obedience to their parents. This ideology may establish parents' dominant position in the family as leaders rather than supporters. When asked about participants' perceptions of their role in the development of young people's mental health, apart from 'crucial', the words 'leading' and 'guiding' were the most frequently repeated answers. For instance:

Parents should guide their children at an early age and help them construct a normal view of life as early as possible *(Female, 49 years old, 22-year-old child, bachelor's degree)*.

Parents play the role of leading and directing them *(Male, 49 years old, 18-year-old child, bachelor's degree)*.

Interestingly, by 'leading', participants often referred to directing young people to a 'normal' path that is distinguished from 'abnormal' one which leads to MHD. However, it should be noticed that some participants held different views:

… parents should be the ones to cure, accompany and encourage their children … Empathy is the key. It's about understanding your children's point of view, not about telling them what to do *(Female, 47 years old, 21-year-old child, high school degree)*.

Meanwhile, the family hierarchy may lead to parents attributing the responsibility of maintaining mental health to young people, despite the perceived leading role of parents. As mentioned above, many participants viewed maintaining good mental health as young people's self-responsibility. It was also believed that it was young people's responsibility to adjust their emotional status and stay strong when faced with stress. Finally, similar responsibility was attributed to the identified difficulties in supporting young people's mental health. The primary difficulty reported was young people's unwillingness to cooperate:

… young people aren't willing to communicate… they are often rebellious towards our preaching or guidance *(Male, 45 years old, 18-year-old child, bachelor's degree)*.

We seldom talk about these things in-depth. He's not patient enough *(Male, 49 years old, 18-year-old child, bachelor's degree)*.

Kids are not at the same level of knowledge or cognitive ability as their parents. Sometimes that leads to different opinions… it's difficult to communicate with them *(Male, 50 years old, 20-year-old child, master's degree or above)*.

Overall, the development, coping, and difficulties in support regarding young people's MHD were mostly perceived as young people's responsibility. On the other hand, parents were perceived as in a relatively higher position of guiding or leading, rather than directly dealing with MHD.

**Emphasis on harmony: Avoid communication to avoid conflicts.** A considerable number of participants identified the lack of communication as both a cause of MHD, as well as a barrier to parental support. According to the data, an important reason for this involved Confucianism's emphasis on harmony within the family. Conflicts were described as arising due to different opinions on the topic of mental health between older and younger generations:

… we tend to start arguing due to different opinions, when I try communicating with him *(Male, 49 years old, 20-year-old child, bachelor's degree)*.

Due to generation gaps, we have quite different life experiences. It's often the case that we cannot achieve agreement after discussions *(Female, 51 years old, 24-year-old child)*.

As a result, parents discussed reducing communication with their children, or perceived it as useless or unnecessary:

> I rarely communicate with him about these things… I don't think we'll be able to directly talk about it *(Male, 44 years old, 19-year-old child, bachelor's degree).*

> I agree that preaching only triggers negative emotions… so I rarely communicate with her about it. I'd rather be her listener when she has something to complain about *(Female, 49 years old, 19-year-old child, master's degree or above).*

Also, this sometimes resulted in young people's concealment of problems, as noticed by some participants:

> Sometimes the child resists communicating with us and is unwilling to talk... *(Female, 46 years old, 18-year-old child, bachelor's degree)*

> The child seems to be on guard when talking with us… once he realises that we are trying to pry about his opinions, he would retreat and become enclosed *(Female, 48 years old, 18-year-old child, master's degree or above).*

The incentive to avoid conflicts often led to both parents' and young people's unwillingness to directly talk about mental health, which may justify most participants' preference of supporting young people with indirect strategies, as shown in theme *Lack of strategies to cope with anomie.*

### Theme 4: Attitudes of parents with and without diagnosed children

This theme developed though comparisons of the attitudes demonstrated by participants who were aware that their children had experienced MHD, and those who perceived their children as having never experienced such difficulties. However, it should be noted that within each type of parental experience, there existed some degree of variability in opinions. The results presented below focus on the most common responses from each type of participant experience.

**Differences in attitudes.** Many participants who reported that their children had no MHD held relatively generalising and negative impressions of young people with MHD. Meanwhile, participants with diagnosed children seemed to have more personal views of such young people:

> They are under a great deal of stress… they are struggling and pretending to be happy *(Female, 54 years old, 24-year-old child, bachelor's degree).*

> They may be more ambitious in terms of academic performance and life outcomes. However, reality often restricts their ability to achieve those goals *(Male, 55 years old, 22-year-old child, bachelor's degree).*

Furthermore, these participants were more likely to shift the causal attribution of MHD from young individuals to family and parents:

> It's mostly because of inappropriate fostering… we were too concerned about whether she was making progress, like grades and academic performance, and overlooked her personal needs *(Male, 45 years old, 20-year-old-child, bachelor's degree).*

> As parents, we did not provide him with sufficient companionship or communication. He could not depend on us… *(Female, 47 years old, 21-year-old child, master's degree or above)*

Also, these parents reported various negative emotions due to their children's conditions such as guilt, fear and sadness. For instance:

… I was so stressed when I knew that she was suffering from mental disorders… I felt guilty and kept thinking about if it was because I wasn't providing proper support for her… After all, I needed help, especially from professionals like psychiatrists *(Male, 55 years old, 22-year-old child, bachelor's degree)*.

On the other hand, parents who reported that they had never employed any strategies to support young people's mental health, as well as those who perceived that there did not exist social stigma about mental health, were all participants that identified their children as not experiencing MHD.

**Similarities in helplessness.** Both parents with and without children diagnosed with MHD reported that they felt unable to understand or empathise with young people who suffer MHD, which was perceived as an important barrier to support:

… I get that she is suffering, but I cannot empathise with her *(Female, 50 years old, 20-year-old child, bachelor's degree)*.

Sometimes we just cannot understand her thoughts *(Female, 49 years old, 18-year-old child, education level not reported)*.

Therefore, parents commonly reported helplessness and uncertainty about how to support young people effectively:

As a parent, it can be difficult to provide professional support and I am afraid that it may cause backfire effects *(Female, 45 years old, 19-year-old child, bachelor's degree)*.

As the older generation, I feel ashamed but must admit that I do not know the appropriate strategies to support young people's mental health *(Female, 54 years old, 24-year-old child, bachelor's degree)*.

## Discussion

This study explored parents' attitudes towards young people's mental health within the Chinese sociocultural context. The results provide a rich description of parents' perceptions of young people's mental health, highlighting how sociocultural factors such as stigma, anomie and Confucianism can contribute to these attitudes and how attitudes in turn shape parental support for mental health in everyday life. Attitudes of parents with and without diagnosed children were compared in order to examine the diversity of parents' perspectives. Compared to other parents, responses of those who had children with identified MHD contained less stereotypes and more in-depth reflections on causal attributions of MHD. However, they reflected similar internalised stigma and helplessness regarding coping strategies, which were attributable to the common sociocultural context.

In line with labelling theory [32], although this study did not give participants any specific definitions of MHD, but simply mentioned this term, it appeared that they mostly translated the concept into a form of categorisation which distinguishes those 'abnormal' young people from the 'normal' ones. MHD thus represented an abstract, socially constructed idea of abnormality, which did not necessarily need a clinical definition. This confirmed Scheff's [34] argument that individuals may be labelled as deviants not because they have done a certain act, but because they are socially stigmatised as deviants. Parents then perceived their role as guiding their children towards the 'normal' path and prevent their children from becoming labelled as 'abnormal'. This ideology was present throughout parents' responses regarding their attitudes and coping strategies.

Regarding the first research question, unexpectedly, the results illustrated a few paradoxical attitudes that parents held about young people's mental health. First, the very term MHD triggered various negative assumptions such as those framing young people as 'fragile', 'dramatic' and 'incompetent', despite the fact that many of them were aware that these are overgeneralising and stigmatising categorisations which potentially lead to unfair treatment of these young people. Like

previous studies' findings [41,42], these 'abnormal' young people were mostly seen as holding certain defective personalities, being mentally weak or having traumatic experiences. Furthermore, this study complements previous findings by demonstrating that the effects of a label may still exist and generate hostility or loss of social status targeted at the stigmatised group, even when the general public becomes aware of the presence and damage of stigma.

According to Billig's theorising of ideological dilemmas [43], individuals may express contrasting or dilemmatic opinions when their personal experiences are different from perceived common sense, or when they need to justify their accounts on different occasions. Another possible explanation for the contrasting accounts may therefore be that when the participants were explicitly asked about stigma, they employed what Radley and Billig [44] call 'public accounts' and gave what they believed were morally acceptable answers to researchers. Meanwhile, by claiming that it was young people's own doing in developing MHD, the participants might be justifying their lack of direct support for young people's mental health, as presented in the previous section. Therefore, the causal attributions that concentrate on young people's responsibility might in fact be the result of parents' evaluation of the pros and cons of being viewed as bad parents and expressing implicit prejudices against younger generations.

Another paradox that emerged from the study was that parents acknowledged that anomie resulted from their high expectations of young people and its threat to young people's mental health, yet showed limited incentive to change such expectations. This echoes previous findings that Chinese parents often expect young people to stay optimistic and manage their emotions well, even when under stress [13]. Possibly, the one-child policy, together with the collectivistic culture which constructs family as the main source of support for individuals, has resulted in many parents having no choice but to rely on their only child to care for them. Hence, while previous studies have identified that parents' high expectations contribute to young people's fear of becoming a disappointment and reduced mental health [1,23,45], they are relatively limited in acknowledging the dilemma that as the competition for education and work opportunities has become increasingly intense, parents *need to* hold such expectations to ensure the family's future. Accusing parents of holding unrealistically high expectations alone would be insufficient to address this dilemma, as those expectations will not easily change when the sociocultural context stays the same. Instead, some have suggested that development of a long-term care system and better facilitation for older adults in China may reduce their dependency on young people [46]. This may in turn reduce anomie among young adults, thus improving their mental health.

In relation to the second research question, Chinese parents' attitudes were shaped by the sociocultural context in various ways. As the literature shows, the traditional Confucianism ideology which places parents in a dominant position in the family is still evident in contemporary Chinese society [14]. This was reflected in parents' responsibility attributions. Parents largely perceived their role as 'leading' or 'guiding', rather than 'supporting' young people. MHD was likely to be seen as young people's self-responsibility of weakness. This corresponds with previous research finding that Chinese parents often stress the importance of increasing young people's resilience, rather than being concerned about their mental health [1]. Therefore, to improve parental support, it may be essential to first transform this traditional parent-child relationship and perceived family roles.

Additionally, the one-child policy was used by many respondents to justify such causal attributions. Being the only child in the family, younger generations were often perceived as spoiled and 'fragile'. It should be noted that these labels represent a unique group of stereotypes *specifically* assigned to these young generations to explain mental health, which may not apply to other groups struggling with similar problems.

Like previous research [10,41], this study found that stigma often extended to negative assumptions of young people's family in China. Participants mostly believed that young people with MHD had unqualified parents, or abnormal families. This is consistent with previous findings that caregivers of patients diagnosed with mental illnesses tend to experience extended and internalised stigma, including embarrassment, shame and discrimination [33,47]. Possibly, some may deny their children's MHD to avoid being affected by courtesy stigma [31,48], by claiming that it is only young people being dramatic or 'fragile'.

This finding connects to the third research question, as the topic of mental health was largely seen as sensitive and was often not directly mentioned in the family, resembling previous findings that Chinese parents are less likely to communicate with children about mental health [9,15]. Indirect support was preferred by parents because it was seen as avoiding potential conflicts due to different opinions, which corresponds with the Confucianist ideology that emphasises harmony within the family [14]. Parents without children experiencing MHD were more likely to illustrate stigma or deny the significance of supporting young people' mental health, while those with diagnosed children tended to reflect more on parental causes of MHD, possibly due to differences in personal contact and in-depth information regarding MHD. However, all parents reported helplessness and a need for professional support, as they felt they were never informed of appropriate supporting strategies. This resembles previous findings that Chinese parents are often unsure of how to best provide support when young people disclose MHD [8,13]. Hence, a fundamental problem may be that specific knowledge about MHD, such as causes and outcomes, remains absent from the public's general knowledge [26]. Parents will likely benefit from enhanced education about the consequences of young people's MHD, as well as effective methods of family support. Overall, parents in the study demonstrated awareness that they should support young people's mental health, yet did not know how to provide such support.

## Limitations and future directions

There are several limitations of this study that need to be acknowledged. Data for this research were collected through an online survey with participants recruited via WeChat and Rednote. Although this method allowed for a relatively large sample, it limited participants to those who had access to the Internet. Additionally, users of these two platforms might hold relatively higher awareness regarding mental health, resulting in selective bias in their report of attitudes and coping strategies. Participants were also predominantly urban residents, possibly due to this selective bias. Therefore, all claims of frequency or differences between groups (e.g., comparing responses from parent with and without children with a diagnosis) are solely intended to describe the sample and the qualitative responses in our dataset and are not intended to be statistically generalizable to the larger population. Due to the nature of a survey rather than interviews, sometimes responses were limited in detail or incomplete. Many responses are therefore relatively brief and show similar patterns in content. Future research may employ more in-depth qualitative methods like one-to-one interviews, which may help explore more subtle attitudes and group differences data in relation to demographic characteristics, such as gender differences between mothers' and fathers' attitudes.

Moreover, this study was solely based on parents' perspectives. Future research may concentrate on young people's perspective and gain insights from both sides of the parent-child relationship. Furthermore, given the aims of the present research were focused on parents of emerging adults, the age of participants' children was therefore restricted to the ages of 18–25. Future research interested in sociocultural factors shaping the attitudes of parents of new cohorts of emerging adults should pay attention to the ways in which generational shifts may alter the role of culture and policy (e.g., one-child policy no longer in effect).

This study concentrated on exploring parents' attitudes towards young people's mental health, including their impressions, causal attributions, perceived stigma and coping strategies. Future research on this topic is needed, given the resulting knowledge will likely prove useful in developing effective interventions to change parents' internalized stigma and improve parents' direct support for young people's mental health. Particularly, as shown by this research, parents' attitudes are often embedded and shaped by the sociocultural context of Chinese society. Thus, multi-level interventions may be required to effectively transform current patterns of family support.

## Conclusion

This study provides preliminary yet important knowledge about Chinese parents' attitudes towards and support for young people's MHD, as well as the ways in which they are shaped by the Chinese sociocultural context. The findings

highlighted that parents' attitudes cannot be separated from other sociocultural factors that are unique to the Chinese set-ting, including traditional patterns of family relationships, social policies, anomie, social stigma and lack of education about mental health. Future research is needed to further document the external factors that have shaped parents' paradoxical attitudes and perceived helplessness regarding children's MHD. Knowledge produced from such research will likely have practical implications for parents' everyday practices of supporting young people in their struggles with MHD.

## Author contributions

**Data curation:** Ziqi Huang.

**Formal analysis:** Ziqi Huang.

**Investigation:** Ziqi Huang.

**Methodology:** Ziqi Huang.

**Supervision:** David M. Frost.

**Writing – original draft:** Ziqi Huang.

**Writing – review & editing:** David M. Frost.

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
