## [Decision Letter · Decision Letter 0]

PMEN-D-24-00463

The 'fragile' generation: Understanding parents' attitudes towards young people's mental health within the Chinese sociocultural context

PLOS Mental Health

Dear Dr. Huang,

Thank you for submitting your manuscript to PLOS Mental Health and we are sorry for the delays experienced. After careful consideration of the reviewer reports, we feel that it has merit but does not fully meet PLOS Mental Health’s publication criteria as it currently stands. Therefore, we invite you to submit a revised version of the manuscript that addresses the points raised during the review process. Please ensure that you fully address all points raised by the reviewers, as we will be unable to send this back to them if the points are not fully addressed. 

We look forward to receiving your revised manuscript.

Kind regards,

Karli Montague-Cardoso

Executive Editor

PLOS Mental Health

Journal Requirements:

https://journals.plos.org/mentalhealth/s/figures 

https://journals.plos.org/mentalhealth/s/figures#loc-file-requirements 

2.  Please provide an Author Summary. This should appear in your manuscript between the Abstract (if applicable) and the Introduction, and should be 150–200 words long. The aim should be to make your findings accessible to a wide audience that includes both scientists and non-scientists. Sample summaries can be found on our website under Submission Guidelines:

https://journals.plos.org/globalpublichealth/s/submission-guidelines#loc-parts-of-a-submission.

3. In the online submission form, you indicated that "Data will be made available on request.". 

a. In a public repository, 

b. Within the manuscript itself, or 

c. Uploaded as supplementary information.

Additional Editor Comments (if provided):

Reviewers' comments:

Reviewer's Responses to Questions

**Comments to the Author**

1. Does this manuscript meet PLOS Mental Health’s publication criteria ? Is the manuscript technically sound, and do the data support the conclusions? The manuscript must describe methodologically and ethically rigorous research with conclusions that are appropriately drawn based on the data presented.

Reviewer #1: No

Reviewer #2: Yes

2. Has the statistical analysis been performed appropriately and rigorously?

Reviewer #1: No

Reviewer #2: Yes

3. Have the authors made all data underlying the findings in their manuscript fully available (please refer to the Data Availability Statement at the start of the manuscript PDF file)?

Reviewer #1: No

Reviewer #2: Yes

4. Is the manuscript presented in an intelligible fashion and written in standard English?

Reviewer #1: No

Reviewer #2: Yes

5. Review Comments to the Author

Reviewer #1: Thanks for providing me the opportunity to review the manuscript. The main concerns are:

1. Some information might mislead international audience. For example, the authors pointed out that “currently a lack of official data about young people's mental health from Chinese institutions”. In fact, there is official dataset but it is not free for the public.

2. Although the authors emphasized that sociocultural might be an important influential factor on young people’s mental health, I do not think that they illustrate what a Confucianist family means clearly. The authors missed many key references in this section.

3. Any ethics approval?

4. The recruitment process is not clear and the authors did not state clearly the rationale for using the method.

5. No questionnaire was provided. Personally, I do not know how the data was analyzed.

6. Personally, I would suggest that the authors do a proof-reading again as many sentences are awkward.

Reviewer #2: The manuscript provides an insightful qualitative exploration of Chinese parents' attitudes towards young people's mental health, contextualized within sociocultural influences such as Confucianism, the one-child policy, and stigma. The thematic analysis is comprehensive, and the authors effectively integrate relevant theoretical frameworks like labeling theory and Bronfenbrenner's ecological systems model. While this study is timely and topic, several concerns in writing and methodologies are worthy to be addressed before publication:

- Introduction section. While the background is informative, the introduction could benefit from a more concise summary of the existing literature, focusing on gaps that this study aims to address, such as to incorporate literature involving into parental coping strategies and generational shifts. Currently, it feels somewhat repetitive and shallow.

- Research Questions. The research questions are relevant, but they could be more sharply defined. For instance, RQ3 ("How do Chinese parents cope with young people's mental health-related problems?") could specify whether the focus is on practical strategies, emotional responses, or both. The logic to bridge background to research question should be smoothed in the Introduction section.

- More concerns occur into the methodologies. One of the biggest methodological concerns is the under-representation in the sample diversity. The sample is heavily urban (96% urban residents). This limits the generalizability of the findings. Despite it has been acknowledged in the Limitation section, this point should be discussed in more detailed.

- Relating to sampling, as far as I know, both WeChat and Little Read Book included more users who posed interests or fundamental knowledge on this topic, which may further confound selective reporting ingredients into this thematic coding. Please further justify this method over alternatives like interviews, especially considering the sensitivity of the topic.

- Coding Process. The coding process is described, but the manuscript would benefit from more detail on how inter-coder reliability was ensured, even if informal (e.g., through discussion and consensus-building between the authors).

- Age constraint. It is unclear why ages of their offspring are limited into 18-25. If in this case, they are all already adults. In the Chinese socioculture, such cohorts are no longer considered as representative for the entire population (e.g., 00 generation, 10 generation). Please justify this design and discuss potential biases introducing by this constraint.

- Role of Gender. The gender dynamics in parental attitudes are not explicitly discussed. Given the cultural context, it would be interesting to analyze whether mothers and fathers exhibit different attitudes or coping strategies. In the Chinese cultures, societal expectation and roles largely vary from gender.

Minor points

- The manuscript would benefit from proofreading to correct minor grammatical issues and improve sentence flow.

- Terms like "mental health difficulties," "mental health issues," and "mental illness" are used interchangeably. It would be helpful to define these terms early in the manuscript and maintain consistency.

6. PLOS authors have the option to publish the peer review history of their article (what does this mean? ). If published, this will include your full peer review and any attached files.

**Do you want your identity to be public for this peer review?** For information about this choice, including consent withdrawal, please see our Privacy Policy .

Reviewer #1: No

Reviewer #2: **Yes: ** Zhiyi Chen

---

## [Decision Letter · Decision Letter 1]

The 'fragile' generation: Understanding parents' attitudes towards young people's mental health within the Chinese sociocultural context

PMEN-D-24-00463R1

Dear Ms. Huang,

We are pleased to inform you that your manuscript 'The 'fragile' generation: Understanding parents' attitudes towards young people's mental health within the Chinese sociocultural context' has been provisionally accepted for publication in PLOS Mental Health.

Best regards,

Karli Montague-Cardoso

Staff Editor

PLOS Mental Health

Reviewer Comments (if any, and for reference):

Reviewer's Responses to Questions

**Comments to the Author**

1. If the authors have adequately addressed your comments raised in a previous round of review and you feel that this manuscript is now acceptable for publication, you may indicate that here to bypass the “Comments to the Author” section, enter your conflict of interest statement in the “Confidential to Editor” section, and submit your "Accept" recommendation.

Reviewer #2: All comments have been addressed

2. Does this manuscript meet PLOS Mental Health’s publication criteria ? Is the manuscript technically sound, and do the data support the conclusions? The manuscript must describe methodologically and ethically rigorous research with conclusions that are appropriately drawn based on the data presented.

Reviewer #2: Yes

3. Has the statistical analysis been performed appropriately and rigorously?

Reviewer #2: Yes

4. Have the authors made all data underlying the findings in their manuscript fully available (please refer to the Data Availability Statement at the start of the manuscript PDF file)?

Reviewer #2: Yes

5. Is the manuscript presented in an intelligible fashion and written in standard English?

Reviewer #2: Yes

6. Review Comments to the Author

Reviewer #2: (No Response)

7. PLOS authors have the option to publish the peer review history of their article (what does this mean? ). If published, this will include your full peer review and any attached files.

**Do you want your identity to be public for this peer review?** For information about this choice, including consent withdrawal, please see our Privacy Policy .

Reviewer #2: No
